# The IHAT-GUT Iron Supplementation Trial in Rural Gambia: Barriers, Facilitators, and Benefits

**DOI:** 10.3390/nu13041140

**Published:** 2021-03-30

**Authors:** Isabella Stelle, Lorraine K. McDonagh, Ilias Hossain, Anastasia Z. Kalea, Dora I. A. Pereira

**Affiliations:** 1Division of Medicine, Institute of Liver and Digestive Health, University College London (UCL), London WC1E 6BT, UK; 2Women and Children’s Health Department, Division of Medicine, King’s College London (KCL), London SE1 7EH, UK; 3Research Department of Primary Care and Population Health, University College London (UCL), London NW3 2PF, UK; l.mcdonagh@ucl.ac.uk; 4Medical Research Council Unit the Gambia at the London School of Hygiene & Tropical Medicine, Banjul, The Gambia; mihossain@mrc.gm (I.H.); diap2@icloud.com (D.I.A.P.); 5Institute of Cardiovascular Science, University College London (UCL), London WC1E 6BT, UK; 6Department of Pathology, University of Cambridge, Cambridge CB2 1QP, UK

**Keywords:** anaemia, iron deficiency anaemia, iron deficiency, micronutrients, iron, clinical trial, nutrition intervention, public health, global health, malnutrition, low resource setting, qualitative

## Abstract

Introduction: In most sub-Saharan African countries iron deficiency anaemia remains highly prevalent in children and this has not changed in the last 25 years. Supplementation with iron hydroxide adipate tartrate (IHAT) was being investigated in anaemic children in a phase two clinical trial (termed IHAT-GUT), conducted at the Medical Research Council Unit the Gambia at the London School of Hygiene and Tropical Medicine (LSHTM) (abbreviated as MRCG hereof). This qualitative study aimed to explore the personal perceptions of the trial staff in relation to conducting a clinical trial in such settings in order to highlight the health system specific needs and strengths in the rural, resource-poor setting of the Upper River Region in the Gambia. Methods: Individual interviews (n = 17) were conducted with local trial staff of the IHAT-GUT trial. Data were analysed using inductive thematic analysis. Results: Potential barriers and facilitators to conducting this clinical trial were identified at the patient, staff, and trial management levels. Several challenges, such as the rural location and cultural context, were identified but noted as not being long-term inhibitors. Participants believed the facilitators and benefits outnumbered the barriers, and included the impact on education and healthcare, the ambitious and knowledgeable locally recruited staff, and the local partnership. Conclusions: While facilitators and barriers were identified to conducting this clinical trial in a rural, resource-poor setting, the overall impact was perceived as beneficial, and this study is a useful example of community involvement and partnership for further health improvement programs. To effectively implement a nutrition intervention, the local health systems and context must be carefully considered through qualitative research beforehand.

## 1. Introduction

### 1.1. Iron Deficiency and the IHAT-GUT Trial in the Gambia

Iron deficiency (ID) poses one of the most significant public health burdens today. At any given moment, more individuals suffer from ID than any other health problem, with an estimated 1.24 billion affected individuals worldwide [1]. ID is associated with multiple pathologies, including anaemia and defective organ function [2]. The prevalence of anaemia is five times higher in low- and middle-income countries (LMICs) than high-income countries, with ~30% of the world’s population, and 43% of 6–59 months old children, being anaemic [1,3]. In Sub-Saharan Africa, 79% of children under six years are anaemic and iron deficiency anaemia (IDA) affects 58% of pre-school children [4]. As such, IDA is the largest international nutritional deficiency disorder and one of the five leading causes of global disease burden [1].

Despite widespread supplementation schemes, ID prevalence has not changed much in LMICs over the last 25 years [5]. There is growing interest in developing novel nano-iron compounds or delivery systems for fortification and supplementation [2,6,7,8,9]. One proposed strategy is a targeted-release nano-iron formulation [10]. Iron hydroxide adipate tartrate (IHAT) and standard-of-care ferrous sulphate were tested in a randomised placebo-controlled double-blind clinical trial (acronym IHAT-GUT) conducted at the Medical Research Council Unit the Gambia at the London School of Hygiene and Tropical Medicine (LSHTM) (abbreviated as MRCG hereof) [11].

### 1.2. Nutrition Interventions in Varying Contexts and the Need for Qualitative Data 

While integrating nutrition-specific interventions into health systems can be impactful for both health and nutrition outcomes, different countries will have specific delivery needs for implementation [12]. Until the barriers and facilitators of nutrition intervention trials are studied across various settings, there will be a lack of data to implement such interventions. Additionally, LMICs remain underrepresented in research [13]. The Global Forum of Health Research termed this the “10/90 gap” to exemplify that less than 10% of health research funds go towards problems affecting 90% of the population worldwide [14], with a smaller percentage towards LMICs [15,16]. Evidently, clinical research is skewed, with more than 80% of clinical trials occurring in high-income countries [17,18,19] and only ~1% of drugs produced between 1975–2004 addressing LMIC issues [20]. Research enhancement in LMICs is an efficient and beneficial way to correct this gap [14].

### 1.3. Clinical Nutrition Trials in LMICs

Given the state of healthcare and disease prevalence in LMICs, clinical trials are well received, and participant recruitment is often easier than in a health secure country [13,21]. The ethical argument remains that medicines targeting conditions highly prevalent in LMICs should be tested in those populations, without transferring data from high-income settings. Interventions often have the largest impact medically in LMICs and benefit from involving local staff [22,23,24].

Difficulties of conducting clinical trials in LMICs stem from limitations in obtaining informed consent, ethical compensation mechanisms, poor health infrastructure, socio-economic and cultural differences [21], and lack of education amongst study participants [25,26]. Additional barriers are limited research governance, funding, logistics, commercial ability, infrastructure, research materials, overall research capacity, and unsupportive administrative and government systems [15,16,25,27,28,29,30]. A recent review looking at existing integrated health and nutrition programmes across 45 LMIC settings found that service delivery and health workforce were well-integrated, but governance, information systems, finance and supplies, and technology were less well-integrated [12].

The aim of this study was to: qualitatively explore IHAT-GUT trial staff perceptions of barriers and facilitators to conducting this clinical trial to highlight the health system specific needs and strengths in a rural and resource-poor setting. 

## 2. Methods

### 2.1. IHAT-GUT

IHAT-GUT was conducted on children with anaemia between the age of 6 and 35 months, living in The Upper River Region (URR) of the Gambia [11]. The children were enrolled in the trial for 113 days, within which they underwent supplementation for 85 days, with weekly study visits to test haemoglobin (Hb) levels and malaria status, and three study timepoints included venous blood collection. Further information about IHAT-GUT study protocol is provided in the protocol paper [11].

### 2.2. Study Setting 

We describe the involvement of research staff in the iron supplementation trial IHAT-GUT [11], which was under the governance of MRCG (Figure 1). The Gambia is the smallest and most densely populated country in West Africa, with about 2.28 million inhabitants, of which roughly one million (48.6%) live below the national poverty line [31]. Islam is the predominant religion, polygamy is widely practiced, and families live in multigenerational compounds within villages [31]. The Gambia is subject to bimodal weather conditions having a “wet” (June to October) and “dry” (November to May) season. The seasonal rain determines the farming practices at that time of year, lending to extreme variations in seasonal diets and fluctuating levels of malnutrition [32].

### 2.3. Study Participants

Individual interviews were conducted with 17 IHAT-GUT local trial staff (Table 1). Trial staff were purposively sampled to ensure insights from varying job types. To maintain anonymity, participants are identified via job title only. 

### 2.4. Data Collection 

Data was collected in person by the first author (IS). Participants chose the location of the interviews to facilitate a more comfortable environment. Locations included the MRCG in Basse and Fajara and study sites within the URR. A topic guide (available upon request) was developed and used to facilitate discussions. Interviews, with informed consent, were audio recorded and later transcribed verbatim. Photographs were taken during data collection with informed consent. Interviews continued until no additional insights were gained (i.e., data saturation [33]).

### 2.5. Data Analysis 

Transcripts were analysed using inductive thematic analysis to explore patterns in the data [34]. Braun and Clarke’s six phases of thematic analysis were used (Table 2) [34]. The transcripts were read several times (data familiarisation). Notes were made in relation to significant and/or interesting comments made by interviewees. The transcripts were coded, and memos written. A code represented a feature of the data that the researcher found interesting or insightful in relation to the research questions, and the memo was a summary of the findings [34]. A list of codes was constructed and connections between them sought to develop provisional themes (repeated patterned responses within data sets) and sub-themes [34]. When all transcripts were analysed, a final list of themes and sub-themes was created. 

An inductive approach was used whereby data analysis was data driven so that participant’s views took precedence over the interviewer’s previous knowledge or beliefs [34]. That said, previous knowledge will, to some extent, influence the research. Therefore, to ensure rigour and validate the finding, transcripts were analysed by the second author (LMD) using the same procedure. Discrepancies were discussed and jointly altered. 

### 2.6. Ethical Considerations

Ethical approval was obtained from the Gambia Government/MRC Joint Ethics Committee (REF: L2018.25) in June 2016. The study was conducted in accordance with the Declaration of Helsinki. Written and verbal informed consent for interviews and photographs was obtained from each participant. It was made clear that refusal to be involved would be confidential and would not affect their work; no participants refused to be interviewed.

## 3. Results

A distinction was made between barriers and facilitators at three levels: study participant, trial staff, and trial management. 

### 3.1. Barriers 

The barriers included community factors and low incentivisation (participant level), motivation (staff level), and country context (trial management level) (Figure 2). Illustrative quotes from each sub-theme of the barriers can be found in Table 3. 

#### 3.1.1. Participants

##### Community Factors 

Many viewed specific cultural factors as barriers to successful running of this trials, specifically community hierarchies and education. These issues posed challenges with community sensitisation for the staff, but it was noted that these barriers were easily overcome if approached properly. Regarding hierarchies, the Local principal investigator (PI) noted that to run the trial fluidly it was key to “*pass the messages [to the communities] smoothly, so everyone can understand. If your communities understand you won’t have problems… with sensitisation go through the hierarchy: the village level, then compound level, then houses, then individuals*”. Another community norm is that mothers often stay home and mind the children and household, but during IHAT-GUT, mothers and children had to attend clinic once a week. The Data Manager stated: “*There’s also a bit of cultural problem… because in Africa we believe the wife stays home to cook, clean. So, some husbands decide (to) have their wives stop going to the clinic visits*”. The issue of husbands not wanting their wives to leave home to attend clinic visits was a barrier to participant recruitment, but with appropriate informed consent and village hierarchical involvement, this was overcome. 

Community education levels were believed to impact the communication of trial information. While many men attend secondary school, women often leave school during puberty to start a family. The low level of education was viewed as a barrier to informing communities about the trial: “*The level of awareness, literacy and education here is a challenge because you want to be sure that the people you are conducting research on are aware of what your study is about and what the implications are*” (Research Clinician). Another barrier was the misunderstanding about blood sampling. State Enrolled Nurse 1 pointed out that some parents were reluctant for their child to join the trial (and others withdrew) when they learnt blood drawing was involved (Figure 3). This was linked to lack of education and local attitudes towards confusion around clinical research: “*Sometimes the awareness is an issue, some people have the wrong sentiment with the local mindset*” (Data Manager). To overcome this barrier, IHAT-GUT used village sensitisation to inform all villagers about the trial and of what it consisted. 

##### Low Incentivisation

State Enrolled Nurse 2 maintained that offering free medication and healthcare was an incentive to take part in the trial but moving forward researchers in other clinical trials need to consider incentives beyond the child’s healthcare: “*When we do the venous bleeding, we give them (the mothers) 50 Dalasis (to buy breakfast), but I would love for it to be more than that. We take blood from these children, so we need to make life easy for them. Maybe they are on the drug that doesn’t do anything. The mothers sacrifice a lot for this project, so we need to give them something back*”. 

#### 3.1.2. Staff

##### Motivation

As summarised by the Research Clinician, there is always challenges when working within teams: “*There will always be challenges. That you should expect. The biggest challenge is working with individuals and managing individuals. Everyone has negative qualities, I have them*”. Workload was also noted as a barrier by both the laboratory and data teams, as those departments “tend to be oversubscribed” (Statistician). The Analytical Project Manager commented that the workload became unmanageable for the laboratory staff at some points. Staff motivation was difficult for those with managerial responsibilities (See Appendix A): “Although this is not the first time, I’ve manned a group like this, it’s not easy” (Field Coordinator; managed thirty employees).

#### 3.1.3. Trial Management

##### Country Context

The remote setting and harsh weather conditions required extensive planning and coordination: “*Resources are always a problem in developing countries*” (Statistician). Likewise, “*IHAT-GUT is running its study where no study has been done in the past at this scale… during rainy season you have floods*” (Nutrition Theme Administrator). The Research Clinician mentioned that: “*The weather… it’s harsh, it’s quite hot, dusty, you get flooding*”. It was also mentioned that: “*You need to consider the environment and circumstance (they) are in a very robust area in the North Banks*” (Local PI) where “*the villages are far, and the roads are poor*” (Senior Field Worker). For example, “*let’s say you’re living in a place like Kuwonkuba, they might not have electricity or clean water. This adds stress*” (State Enrolled Nurse 1).

The Nutrition Theme Administrator noted that the context made sticking to the tight project timelines challenging: “*The major challenges in running these trials here is… making all the necessary arrangements: approvals from the Ministry, Medicine Control Agency, Ethics and SCC (Scientific Coordinating Committee), establishing the sites of the studies. And of course, it requires a lot of logistical support in remote areas*”. He also noted handling the “*financial management*” of various projects at MRCG in relation to context: “*Always expect a project to overrun, so you adequately allocate expenditures. For example, in clinical trials, you always have SAEs (Serious Adverse Events)*”. The associated expenditures due to the context also posed a challenge for the Field Coordinator: “*It’s a problem… we are spending too much on the (ferry) crossing*”. Likewise, transporting mothers from villages to clinics and transferring samples to the main laboratory facilities added financial stress. The Senior Field Worker noted funding needs such as giving “*(phone) credit to call for the Field Workers and mothers*” so they could afford to make trial related calls.

### 3.2. Facilitators & Benefits

The facilitators included healthcare, incentivisation, and receptive communities (participant level); staff characteristics and education enhancement (staff level); and the local partnership (trial management level) (Figure 4). Illustrative quotes from each sub-theme of the facilitators can be found in Table 4.

#### 3.2.1. Participants

##### Healthcare

The study was conducted in the North Bank of the URR where the poorest Gambian communities reside and the double burden of malnutrition and infection is highest [35]. Scientific Officer 2 noted: “*It’s better to give nutritional interventions where they are most needed*”. This was echoed by the Senior Field Worker: “*Especially IHAT-GUT because it helps the Gambian children and it’s the first one in the North Bank… We are working where they need it most...There was more anaemia, they had less hospitals. Medical care is lacking. Mothers say: ‘please come to our communities’*”.

Another advantage for trial participants was that healthcare was fully covered during the trial. The Nurse Coordinator found that local communities embraced these interventions: “*These studies excite them. They want them to continue. We deal with the health, the management, the transportation.*”

##### Incentivisation

The Data Manager mentioned that: “*You need to bring a social impact, so the participants feel valued (rather) than just coming to do what you want and not giving the mothers and children something*”. A successful incentive was the provision of Yandi juice, which made supplement administration easier and more enjoyable for participants: “*Giving the Yandi attracts the children because they want to take the drugs. When they see the Field Workers, they get excited*” (State Enrolled Nurse 2). The Nurse Coordinator also mentioned how “*sometimes the kids are running for the Fields Workers because they are excited for the juice*” and advised to use this approach in the future.

##### Receptive Communities

Staff found the communities were receptive towards the study. The Research Clinician, who is not Gambian himself, found: “*The Gambians, they are remarkable people. They are the most amazing, welcoming people. It’s a very friendly environment to work in. It’s a research-friendly country*”. State Enrolled Nurse 2 highlighted that URR, being a low resource setting, facilitated this: “*The advantage of the project being run here is due to low-income earners, so having these projects is a big deal… They are very cooperative. In an urban area it would have been harder, but in a rural area they are excited*”. The Data Manager echoed this sentiment and believed a similar trial in the UK would be more challenging. 

#### 3.2.2. Staff

##### Staff Characteristics 

It was very helpful that the staff going into the communities to conduct research were from a similar cultural background and could speak local languages. The Senior Field Worker highlighted: “*In Europe, I don’t think it would be easy to conduct studies like this. In Africa, people don’t find it a problem that projects come in their communities. We are Gambian. When we go into our own communities, they are accepting*”.

The staff had strong experience and knowledge in working with research studies in rural Gambia. The Local Safety Monitor was confident that being part of this project meant having a high calibre of staff with education, knowledge, and experience: “*The teams, the doctors, the researchers, the nurses and so on that we have here are second to no other country, so you know they can get good work here*”. The Analytical Project Manager had the same confidence in his colleagues: “*It’s also that they have people on ground who already have experience of running these clinical trials, both from the clinical aspects to the field staff. The people are experienced and consistent* (…) *they have had proper training about interacting with the community and attaining data from them*”. 

Staff were highly ambitious for themselves and the development of their country. The Local Safety Monitor mentioned impacting West Africa through public health: “*(It’s) very important to me because it drives you to continue your work, you see the issues and the results and it’s inspiring to keep doing this work*”. Having grown up in the Gambia, the Data Manager was proud to impact his country: “*Growing up here, I’ve always seen malnutrition and poverty, so I’m conscious of my country’s state* (…) *you are contributing to change young people’s lives, which is just amazing*”. Likewise, the Senior Field Worker found: “… *I can contribute my part to help Gambian children*”.

##### Education Enhancement 

The staff felt valued by MRCG and IHAT-GUT, thanks to the knowledge that they gained and this enriched their work experience and dedication to the trials (See Appendix A). The Field Worker was proud of his personal learning and was grateful for the educational opportunities: “*One thing I like in IHAT-GUT is the blood experience because it’s a learning experience*”. One of The Statistician’s favourite parts of working with MRCG was their advancement of staff knowledge through training. Likewise, Scientific Officer 2 was able to complete her master’s because of MRCG. 

#### 3.2.3. Trial Management 

##### Local Partnership 

MRCG’s research unit has been in the Gambia for 70 years, giving them respect in the country for their long-standing establishment, advances in healthcare, and facilities for epidemiological studies and clinical trials. MRCG’s upstanding reputation with The Gambian Government and communities allowed easier collaboration and authorisation from the government: “*MRC has a great track record here in the Gambia, they have cordial relationships with the communities and with The Gambian government and its ministries*” (Nutrition Theme Administrator). The Field Coordinator noted that MRCG has an international research reputation: “*They have a high regard in terms of quality research, and they have been in existence for almost 70 years*” (See Appendix A). The Analytical Project Manager was grateful for the high-quality facilities, which allowed the project to run smoothly (Figure 5). 

## 4. Discussion

Clinical research in a LMIC, such as the Gambia, poses challenges. Barriers emerged in this study, such as mistrust regarding blood drawing. O’Neill et al. [26] looked at the misconceptions around blood and its impacts on trial participation in a village in rural Gambia. Originally, participant recruitment for finger-pricking for posed no problem, but as soon as rumours about the use of the blood and concerns around the health implications of blood loss started spreading, only 42% of the inhabitants consented [26]. They concluded that to overcome this barrier one must better inform and educate the villagers [26]. Likewise, as noted in the literature and concurrent with our findings, in order to facilitate clinical trials, health education is crucial to avoid misconceptions about diseases in the Gambia [25].

While LMICs were noted as lacking commercial ability, infrastructure, materials, and overall research capacity, elsewhere, this was not seen in the present study [15,16,27]. Running clinical trials through MRCG appeared as a facilitator, while unsupportive administrative systems were reported to be a barrier for clinical trials in LMICs [27,28]. In terms of “human capacity,” a recurrent theme in this data was the experienced MRCG staff and investment in training. In contrast, Ross et al. [33] found that in developed countries, such as the United States, United Kingdom, Europe, and Australia, lack of staff and adequate training posed barriers. Likewise, in our study staff motivation was noted as occasionally posing a barrier, which is concurrent with the literature [27,36].

Materials can be difficult to resource in LMICs and for projects to run smoothly; advanced planning is needed. However, a lack of resources in LMICs increases the need for sound research to prioritise these limitations [37]. This barrier of planning and preparation not only encompasses the materials needed, but also appropriate government approvals, which was previously noted [27,36]. MRCG’s long-standing establishment and good government relations ease this potential barrier due to expediated government and ethical approvals as well as familiarity with product procurement in a resource-limited setting. Overall, the participants showed a clear awareness of the country contexts’ barriers and facilitators, and the implications these have for the trial. Going forward, building research capacity by conducting more clinical trials in LMICs is vital to ease the burden of what are already rigorous and time-consuming study preparation phases, only complicated by the resource limitations of LMICs.

The overall impact for the country’s healthcare can be beneficial and outweighs the barriers described. Our findings are in line with previous research, noting the need for research in such communities [13,21]. Clinical interventions in LMICs have the largest impact in decreasing childhood mortality rates [13,21]. The high infection burden setting of rural Gambia was the ideal setting for IHAT-GUT. If the drug were to be tested in a high-income, low-infection burden country, its effects might not be translatable to the country of target, as seen in other clinical trials [38,39].

The ease in recruiting patients was noted as a facilitator in running clinical trials in LMICs like the Gambia. LMICs offer an attractive setting for clinical trials as there is often less access to healthcare, so the prospect of healthcare through a clinical trial lends to shorter periods of participant recruitment [21]. Recruitment time can be five to ten times quicker in LMICs than in developed regions such as the United States or Europe [30]. For example, one participant highlighted that he did not think mothers in developed countries would be happy to enrol their child in a study, but in the Gambia the mothers are willing. Ross et al. [33] similarly reported in his systematic review of 78 randomized control trials across the United States, United Kingdom, Europe, and Australia that recruitment for clinical trials was a challenge. Overall, the staff’s observations about the trial setting being a facilitator for participant incentivisation and ease of recruitment due to increased access to healthcare is in agreement with previous literature.

Thanks to the local partnership and strong establishment of MRCG in the Gambia, the country is well equipped for incoming projects due to receptive communities, ambitious and knowledgeable locally recruited staff, and the research facilities and governance offered by MRCG. Locally recruited staff working in their own communities made communities receptive to the trial due to increased trust. It was previously reported that it was beneficial for clinical trials to utilise their local workforce because it allowed for the use of local knowledge [22]. This made the trials more responsive to the country’s needs and more effective in influencing policy [24]. Likewise, the staff were aware that passing knowledge through the family hierarchies may facilitate participant recruitment. Preliminary community sensitisation that allowed information to be passed through the appropriate village hierarchy in the Gambia has been successful in the past [23].

Another highlighted facilitator in the trial, beyond just participant healthcare, was participant transportation to and from the clinic where the interventions took place. Mobility has been seen as a barrier for non-participation in a clinical trial in the Gambia [25,29].

## 5. Limitations

A few limitations of this study warrant noting. Data was collected by a young white woman interviewing mostly older men. Broom et al. [40] found heightened “professionalism” and self-credentialing by men when interviewed by a woman. Likewise, it has been noted that gender in research warrants more attention, especially in the context of women interviewing men, such as men’s assertions of gender identities and gender hierarchy [41]. Additionally, the PI (IS) worked closely with the PI of IHAT-GUT (DIAP) and an association of the two women being colleagues may have influenced participant answers due to concern around socially desirable responding in high-stakes situations where participants aim to make a good impression [42]. Lastly, IHAT-GUT was noted to be well run, using different monitoring, training and consenting systems that have not been used by a study in those communities before. Therefore, facilitators may have been more prominent [27], making it harder to relate to other clinical trials.

## 6. Conclusions

This study highlighted the barriers and facilitators to conducting this clinical trial in a rural and resource-poor setting. It brought to light that for clinical trials to be successful in such settings, cultural context must be carefully considered. Specifically, researchers should devote substantial time to engaging with the community to gain insight into pre-existing beliefs, knowledge, and awareness levels of the population, as well as the social structures at play.

This study reported that the staff were proud of their high calibre of work and ambitious to continue making an impact on the country’s education and healthcare levels. While barriers were faced in running a clinical trial in this rural, resource-poor setting, the overall impact was perceived as beneficial, and this study is a useful example of community involvement and partnership for further health improvement programs. These findings highlight the value that staff find clinical trials add to their lives as well as the need for creating and nurturing local partnerships, which enables the continuous embrace and success of clinical research.

## Figures and Tables

**Figure 1 nutrients-13-01140-f001:**
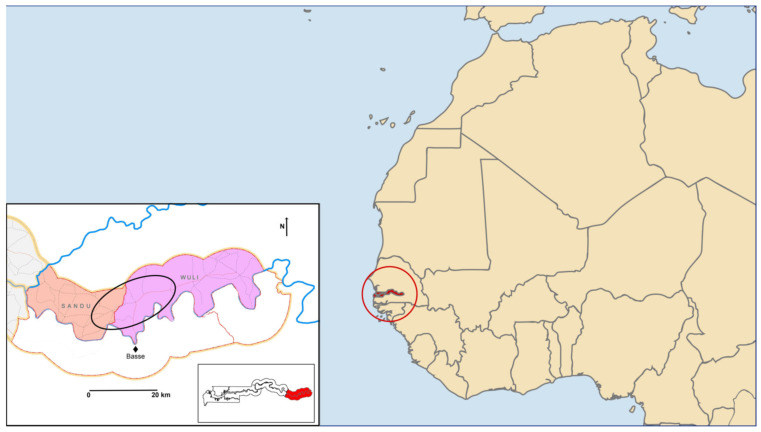
Study catchment area (circled in black) in the Upper River Region (red inset) of the Gambia (circled in red) in West Africa (main picture).

**Figure 2 nutrients-13-01140-f002:**
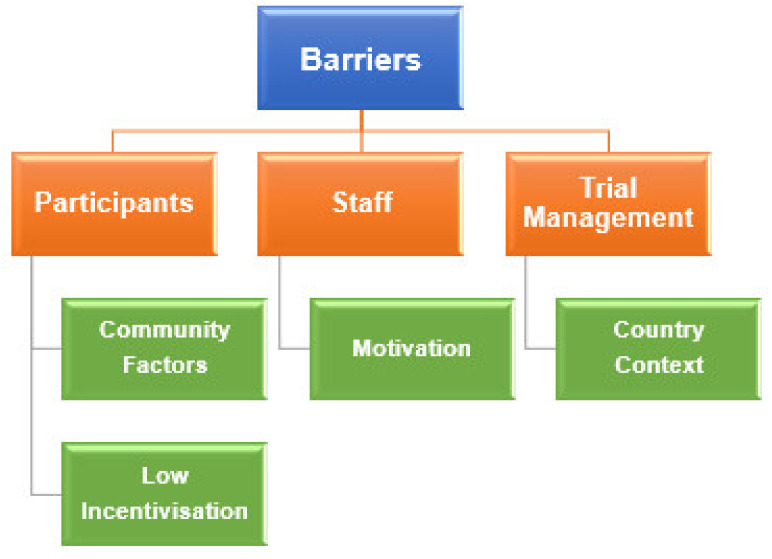
Graphical illustration of themes and sub-themes of the barriers.

**Figure 3 nutrients-13-01140-f003:**
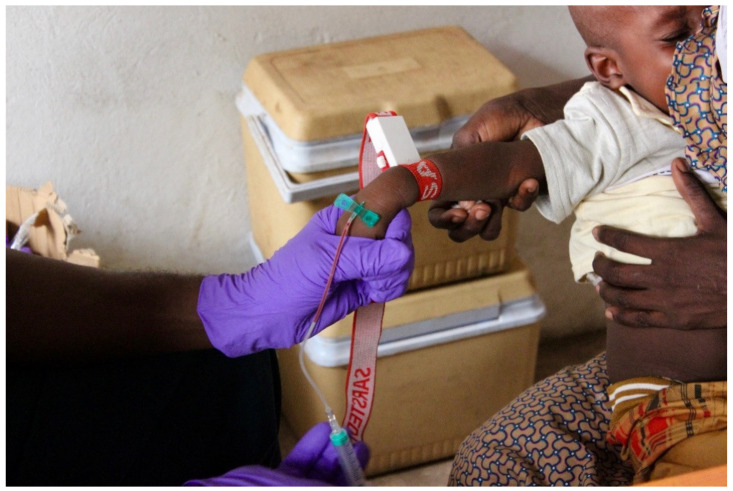
Photograph of blood drawing during a visit to the study clinic.

**Figure 4 nutrients-13-01140-f004:**
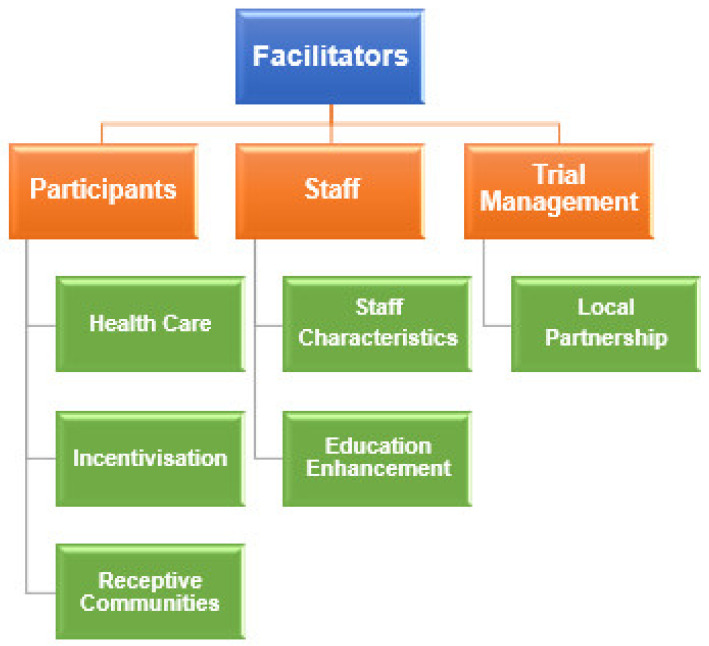
Graphical illustration of themes and sub-themes of the facilitators.

**Figure 5 nutrients-13-01140-f005:**
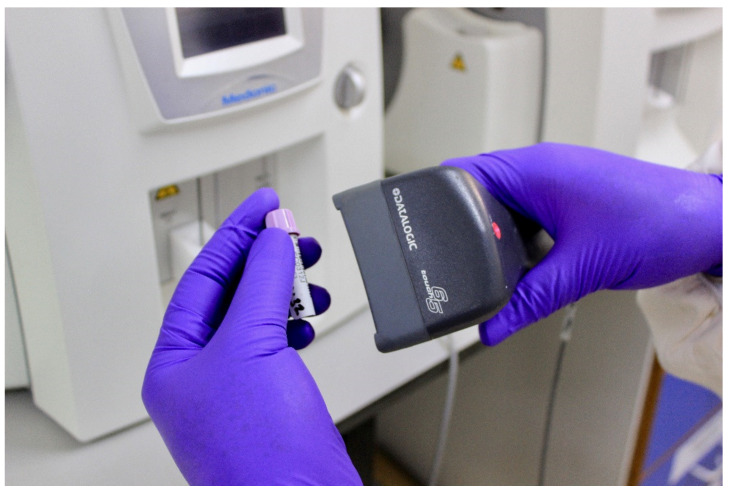
Photograph of sample processing done to Good Clinical Lab Practice standards.

**Table 1 nutrients-13-01140-t001:** Demographic characteristics of sample.

Demographics	n	%
**Ethnicity (*Tribe*) if applicable**		
Gambian (Fula: Mandinka: Wolof: Banbara: Manjago)	12 (6:3:1:1:1)	70
Other African Countries	4	24
Other	1	6
Religion		
Muslim	16	94
Christian	1	6
Sex		
Female	1	17
Male	16	83
Age (in years)		
18–29	1	17
30–39	6	26
40–49	7	30
50–59	3	13
Highest Level of Education		
Secondary School	5	29
State Enrolled Nursing School	3	18
Bachelors	1	5
Medicine Degree	3	18
Masters	3	18
Doctorate ^a^	2	12
Years Employed with MRCG ^b^ Projects		
0–5	3	18
6–10	6	35
11–15	3	18
16–20	4	23
21–25	0	0
25–30	1	6

^a^ = one in progress. ^b^ = MRC Unit the Gambia

**Table 2 nutrients-13-01140-t002:** Six phases of thematic analysis adapted from Braun and Clarke (2006).

Phase	Description
1. Data Familiarization	Transcribing, reading, and re-reading data
2. Initial Codes	Coding interesting features systematically and collating the data to each code
3. Theme Development	Collating codes into potential themes and adding relevant data to each
4. Refining Themes	Ensuring themes work with the first (data familiarization) and second (initial codes) levels of analysis
5. Naming Themes	Ongoing refinement, generating clear definitions and names for each theme
6. The Report	Final analysis opportunity, extraction of compelling examples

**Table 3 nutrients-13-01140-t003:** Illustrative quotes for each sub-theme of the barriers.

Theme	Sub-Theme	Illustrative Quote
**Participants**	Community Factors	“*There’s also a bit of cultural problem… because in Africa we believe the wife stays home to cook, clean. So, some husbands decide (to) have their wives stop going to the clinic visits*.”—Data Manager
	Low Incentivisation	“*We take blood from these children, so we need to make life easy for them. Maybe they are on the drug that doesn’t do anything*.”—State Enrolled Nurse 2
**Staff**	Motivation	“*There will always be challenges. That you should expect. The biggest challenge is working with individuals and managing individuals. Everyone has negative qualities, I have them*.”—Research Clinician
**Trial Management**	Country Context	“*The major challenges in running these trials here is the start up. Making all the necessary arrangements. The necessary approvals from the Ministry, from the Medicine Control Agency, from the Ethics and SCC (Scientific Coordinating Committee), establishing the sites of the studies and so forth. And of course, it requires a lot of logistical support… especially if they are in remote areas. IHAT-GUT is running its study where no study has been done in the past at this scale…(and) during rainy season you have floods*.”—Project Manager

**Table 4 nutrients-13-01140-t004:** Illustrative quotes for each sub-theme of the facilitators.

Theme	Sub-Theme	Illustrative Quote
Participants	Health Care	“*IHAT-GUT…helps the Gambian children and it’s the first one in the North Bank… We are working where they need it most… There was more anaemia, they had less hospitals. Medical care is lacking. Mothers say: ‘please come to our communities’*.”—Senior Field Worker
	Incentivisation	“*You need to bring a social impact, so the participants feel valued (rather) than just coming to do what you want and not giving the mothers and children something*.”—Data Manager
	Receptive Communities	“*The Gambians, they are remarkable people. They are the most amazing, welcoming people. It’s a very friendly environment to work in. It’s a research-friendly country*.”—Research Clinician
Staff	Staff Characteristics	“*In Europe, I don’t think it would be easy to conduct studies like this. In Africa, people don’t find it a problem that projects come in their communities. We are Gambian. When we go into our own communities, they are accepting*.”—Senior Field Worker
	Education Enhancement	“*The Nutritional Course was great. It added value because it not only taught us about nutrition personally, but on the other hand, it’s great to do a team activity. It makes everyone feel appreciated… You want to develop the staff*.”—Data Manager
Trial Management	Local Partnership	“*MRC has a great track record here in the Gambia, they have cordial relationships with the communities and with The Gambian government and its ministries*.”—Nutrition Theme Administrator

## Data Availability

All relevant data are reported within the manuscript.

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
