# Peer review of "The IHAT-GUT Iron Supplementation Trial in Rural Gambia: Barriers, Facilitators, and Benefits"

_nutrients, 2021, doi:10.3390/nu13041140_

Round 1
Reviewer 1 Report
Excellent work
Authors present a very interesting trial and the manuscript is emphasized in barriers concerning trials in developing countries and how beneficial is for inhabitants the presence of such trials. In my opinion it is very important to emphasize in literature about the importance and challenge of such trials and scientists may find the article very interesting.
Author Response
Many thanks for the positive feedback. There are no responses to the reviewer's comments required.
Reviewer 2 Report
This study look at an essential research question - how to enable high quality research in low and middle income countries. The authors are to be commended for reminding us that research on barriers and facilitators should be specific to the area in which an interventional trial/research study is to be conducted, and that more interventional trials should be conducted in the place where the intervention is to be carried out.
Overall I think the analysis is sound and the manuscript well written.
There are some minor grammatical errors that the authors need to review.
My only significant suggestion is in regards to the discussion - in many places discussion is limited and no actual conclusion drawn - the discussion, in parts, thus feels unfinished or superficial - following (after my point about the figures)are places where I think this occurs.
The conclusion also needs reflection - see my later comments
My other query relates to the figures - I found them challenging to interpret. For example, Figure 4 - staff - education enhancement - what does this mean? That if staff are educated they are facilitators? If staff educate participants they are facilitators? Please clarify
Specific points
In the introduction the authors mention a recent review related to service delivery - what kind of review? where?
In data analysis the authors write data that the researcher found "interesting" was analysed - what does that mean? is the data analysis thus biased by personal interests rather than thematic analysis - in qualitative research it is the themes that emerge rather than "interesting data" that is analysed?
In the paragraph about participants I am not sure what the overall conclusion is - please clarify
Section 3.1.3 - country context - what does "excess" mean as used here?
Second paragraph of discussion - can the authors explain the differences and add a concluding sentence providing high level synthesis of the findings
in the paragraph about "ease of recruiting" (BTW - it is so much easier to review if the authors number the lines in their manuscript) in the discussion what is "the issue"?
In the same paragraph - Can the authors explain the differences in studies and provide an over arching conclusion to the paragraph.
The conclusion, to me, feels overly long - in addition, new elements are introduced - that don't belong in a conclusion but rather in the discussion - for example, traansportation
Again, as in some paragraphs in the discussion - the conclusion, succinctly, needs to provide a high level synthesis of presented findings and discussion and implications.
Round 2
Reviewer 2 Report
The authors have addressed all the issues raised.
One final point - please, wherever they refer to a systematic review, can they specify the study type - epidemiological, interventional, qualitative etc
i.e systematic review of 30 interventional studies rather than just systematic review of 30 studies - if mixed then specify study types in mix
Author Response
Two systematic reviews which are referenced have been clarified for study types:
- Review looking at existing integrated health and nutrition programmes across 45 LMIC settings (Salam, Das and Bhutta)
- 78 RCTs across the US, UK, Europe and Australia (Ross et al)
